# ROBUST LEARNING WITH JACOBIAN REGULARIZATION

## ABSTRACT

Design of reliable systems must guarantee stability against input perturbations. In machine learning, such guarantee entails preventing overfitting and ensuring robustness of models against corruption of input data. In order to maximize stability, we analyze and develop a computationally efficient implementation of Jacobian regularization that increases classification margins of neural networks. The stabilizing effect of the Jacobian regularizer leads to significant improvements in robustness, as measured against both random and adversarial input perturbations, without severely degrading generalization properties on clean data.

## 1 INTRODUCTION

Stability analysis lies at the heart of many scientific and engineering disciplines. In an unstable system, infinitesimal perturbations amplify and have substantial impacts on the performance of the system. It is especially critical to perform a thorough stability analysis on complex engineered systems deployed in practice, or else what may seem like innocuous perturbations can lead to catastrophic consequences such as the Tacoma Narrows Bridge collapse (Amman et al., 1941) and the Space Shuttle *Challenger* disaster (Feynman and Leighton, 2001). As a rule of thumb, well-engineered systems should be robust against any input shifts – expected or unexpected.

Most models in machine learning are complex nonlinear systems and thus no exception to this rule. For instance, a reliable model must withstand shifts from training data to unseen test data, bridging the so-called generalization gap. This problem is severe especially when training data are strongly biased with respect to test data, as in domain-adaptation tasks, or when only sparse sampling of a true underlying distribution is available, as in few-shot learning. Any instability in the system can further be exploited by adversaries to render trained models utterly useless (Szegedy et al., 2013; Goodfellow et al., 2014; Moosavi-Dezfooli et al., 2016; Papernot et al., 2016a; Kurakin et al., 2016; Madry et al., 2017; Carlini and Wagner, 2017; Gilmer et al., 2018). It is thus of utmost importance to ensure that models be stable against perturbations in the input space.

Various regularization schemes have been proposed to improve the stability of models. For linear classifiers and support vector machines (Cortes and Vapnik, 1995), this goal is attained via an $L^2$ regularization which maximizes classification margins and reduces overfitting to the training data. This regularization technique has been widely used for neural networks as well and shown to promote generalization (Hinton, 1987; Krogh and Hertz, 1992; Zhang et al., 2018). However, it remains unclear whether or not $L^2$ regularization increases classification margins and stability of a network, especially for deep architectures with intertwining nonlinearity.

In this paper, we suggest ensuring robustness of nonlinear models via a Jacobian regularization scheme. We illustrate the intuition behind our regularization approach by visualizing the classification margins of a simple MNIST digit classifier in Figure 1 (see Appendix A for more). Decision cells of a neural network, trained without regularization, are very rugged and can be unpredictably unstable (Figure 1a). On average, $L^2$ regularization smooths out these rugged boundaries but does not necessarily increase the size of decision cells, i.e., does not increase classification margins (Figure 1b). In contrast, Jacobian regularization pushes decision boundaries farther away from each training data point, enlarging decision cells and reducing instability (Figure 1c).

The goal of the paper is to promote Jacobian regularization as a generic scheme for increasing robustness while also being agnostic to the architecture, domain, or task to which it is applied. In

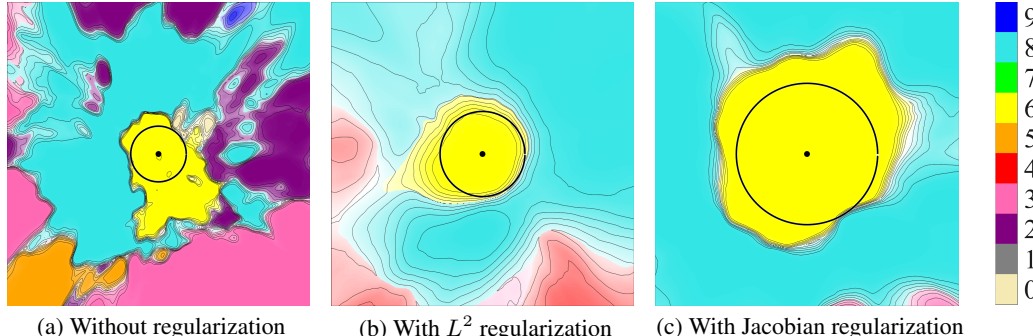

|     | (a) Without regularization | (b) With $L^2$ regularization | (c) With Jacobian regularization |
|-----|---|---|---|

Figure 1: **Cross sections of decision cells in the input space.** To make these cross sections for LeNet' models trained on the MNIST dataset, a test sample (black dot) and a two-dimensional hyperplane $\subset \mathbb{R}^{784}$ passing through it are randomly chosen. Different colors indicate the different classes predicted by these models, transparency and contours are set by maximum of the softmax values, and the circle around the test sample signifies distance to the closest decision boundary in the plane. (a) Decision cells are rugged without regularization. (b) Training with $L^2$ regularization leads to smoother decision cells, but does not necessarily ensure large cells. (c) Jacobian regularization pushes boundaries outwards and embiggens decision cells.

support of this, after presenting the Jacobian regularizer, we evaluate its effect both in isolation as well as in combination with multiple existing approaches that are intended to promote robustness and generalization. Our intention is to showcase the ease of use and complimentary nature of our proposed regularization. Domain experts in each field should be able to quickly incorporate our regularizer into their learning pipeline as a simple way of improving the performance of their state-of-the-art system.

The rest of the paper is structured as follows. In Section 2 we motivate the usage of Jacobian regularization and develop a computationally efficient algorithm for its implementation. Next, the effectiveness of this regularizer is empirically studied in Section 3. As regularlizers constrain the learning problem, we first verify that the introduction of our regularizer does not adversely affect learning in the case when input data remain unperturbed. Robustness against both random and adversarial perturbations is then evaluated and shown to receive significant improvements from the Jacobian regularizer. We contrast our work with the literature in Section 4 and conclude in Section 5.

## 2 METHOD

Here we introduce a scheme for minimizing the norm of an input-output Jacobian matrix as a technique for regularizing learning with stochastic gradient descent (SGD). We begin by formally defining the input-output Jacobian and then explain an efficient algorithm for computing the Jacobian regularizer using standard machine learning frameworks.

### 2.1 STABILITY ANALYSIS AND INPUT-OUTPUT JACOBIAN

Let us consider the set of classification functions, $\boldsymbol{f}$, which take a vectorized sensory signal, $\boldsymbol{x} \in \mathbb{R}^I$, as input and outputs a score vector, $\boldsymbol{z} = \boldsymbol{f}(\boldsymbol{x}) \in \mathbb{R}^C$, where each element, $z_c$, is associated with likelihood that the input is from category, $c$.[1] In this work, we focus on learning this classification function as a neural network with model parameters $\boldsymbol{\theta}$, though our findings should generalize to any parameterized function. Our goal is to learn the model parameters that minimize the classification objective on the available training data while also being stable against perturbations in the input space so as to increase classification margins.

---

[1]Throughout the paper, the vector $\boldsymbol{z}$ denotes the logit before applying a softmax layer. The probabilistic output of the softmax $p_c$ relates to $z_c$ via $p_c \equiv \frac{e^{z_c/T}}{\sum_{c'} e^{z_{c'}/T}}$ with temperature $T$, typically set to unity.

The input-output Jacobian matrix naturally emerges in the stability analysis of the model predictions against input perturbations. Let us consider a small perturbation vector, $\epsilon \in \mathbb{R}^I$, of the same dimension as the input. For a perturbed input $\widetilde{\boldsymbol{x}} = \boldsymbol{x} + \epsilon$, the corresponding output values shift to

$$\widetilde{z}_c = f_c(\boldsymbol{x} + \epsilon) = f_c(\boldsymbol{x}) + \sum_{i=1}^{I} \epsilon_i \cdot \frac{\partial f_c}{\partial x_i}(\boldsymbol{x}) + O(\epsilon^2) = z_c + \sum_{i=1}^{I} J_{c;i}(\boldsymbol{x}) \cdot \epsilon_i + O(\epsilon^2) \quad (1)$$

where in the second equality the function was Taylor-expanded with respect to the input perturbation $\epsilon$ and in the third equality the input-output Jacobian matrix,

$$J_{c;i}(\boldsymbol{x}) \equiv \frac{\partial f_c}{\partial x_i}(\boldsymbol{x}), \quad (2)$$

was introduced. As the function $\boldsymbol{f}$ is typically almost everywhere analytic, for sufficiently small perturbations $\epsilon$ the higher-order terms can be neglected and the stability of the prediction is governed by the input-output Jacobian.

## 2.2 ROBUSTNESS THROUGH INPUT-OUTPUT JACOBIAN MINIMIZATION

From Equation (1), it is straightforward to see that the larger the components of the Jacobian are, the more unstable the model prediction is with respect to input perturbations. A natural way to reduce this instability then is to decrease the magnitude for each component of the Jacobian matrix, which can be realized by minimizing the square of the Frobenius norm of the input-output Jacobian,[2]

$$||J(\boldsymbol{x})||_{\mathrm{F}}^2 \equiv \left\{ \sum_{i,c} \left[ J_{c;i}(\boldsymbol{x}) \right]^2 \right\}. \quad (3)$$

For linear models, this reduces exactly to $L^2$ regularization that increases classification margins of these models. For nonlinear models, however, Jacobian regularization does not equate to $L^2$ regularization, and we expect these schemes to affect models differently. In particular, predictions made by models trained with the Jacobian regularization do not vary much as inputs get perturbed and hence decision cells enlarge on average. This increase in stability granted by the Jacobian regularization is visualized in Figure 1, which depicts a cross section of the decision cells for the MNIST digit classification problem using a nonlinear neural network (LeCun et al., 1998).

The Jacobian regularizer in Equation (3) can be combined with any loss objective used for training parameterized models. Concretely, consider a supervised learning problem modeled by a neural network and optimized with SGD. At each iteration, a mini-batch $\mathcal{B}$ consists of a set of labeled examples, $\{\boldsymbol{x}^\alpha, \boldsymbol{y}^\alpha\}_{\alpha \in \mathcal{B}}$, and a supervised loss function, $\mathcal{L}_{\mathrm{super}}$, is optimized possibly together with some other regularizer $\mathcal{R}(\boldsymbol{\theta})$ – such as $L^2$ regularizer $\frac{\lambda_{\mathrm{WD}}}{2}\boldsymbol{\theta}^2$ – over the function parameter space, by minimizing the following bare loss function

$$\mathcal{L}_{\mathrm{bare}}\left(\{\boldsymbol{x}^\alpha, \boldsymbol{y}^\alpha\}_{\alpha \in \mathcal{B}}; \boldsymbol{\theta}\right) = \frac{1}{|\mathcal{B}|} \sum_{\alpha \in \mathcal{B}} \mathcal{L}_{\mathrm{super}}\left[\boldsymbol{f}(\boldsymbol{x}^\alpha); \boldsymbol{y}^\alpha\right] + \mathcal{R}(\boldsymbol{\theta}). \quad (4)$$

To integrate our Jacobian regularizer into training, one instead optimizes the following joint loss

$$\mathcal{L}_{\mathrm{joint}}^{\mathcal{B}}(\boldsymbol{\theta}) = \mathcal{L}_{\mathrm{bare}}(\{\boldsymbol{x}^\alpha, \boldsymbol{y}^\alpha\}_{\alpha \in \mathcal{B}}; \boldsymbol{\theta}) + \frac{\lambda_{\mathrm{JR}}}{2} \left[ \frac{1}{|\mathcal{B}|} \sum_{\alpha \in \mathcal{B}} ||J(\boldsymbol{x}^\alpha)||_{\mathrm{F}}^2 \right], \quad (5)$$

where $\lambda_{\mathrm{JR}}$ is a hyperparameter that determines the relative importance of the Jacobian regularizer. By minimizing this joint loss with sufficient training data and a properly chosen $\lambda_{\mathrm{JR}}$, we expect models to learn both correctly and robustly.

---

[2]Minimizing the Frobenius norm will also reduce the $L^1$-norm, since these norms satisfy the inequalities $||J(\boldsymbol{x})||_{\mathrm{F}} \leq \sum_{i,c} |J_{c;i}(\boldsymbol{x})| \leq \sqrt{IC}||J(\boldsymbol{x})||_{\mathrm{F}}$. We prefer to minimize the Frobenius norm over the $L^1$-norm because the ability to express the former as a trace leads to an efficient algorithm [see Equations (6) through (8)].

## 2.3 Efficient Approximate Algorithm

In the previous section we have argued for minimizing the Frobenius norm of the input-output Jacobian to improve robustness during learning. The main question that follows is how to efficiently compute and implement this regularizer in such a way that its optimization can seamlessly be incorporated into any existing learning paradigm. Recently, Sokolić et al. (2017) also explored the idea of regularizing the Jacobian matrix during learning, but only provided an inefficient algorithm requiring an increase in computational cost that scales linearly with the number of output classes, $C$, compared to the bare optimization problem (see explanation below). In practice, such an overhead will be prohibitively expensive for many large-scale learning problems, e.g. ImageNet classification has $C = 1000$ target classes (Deng et al., 2009). (Our scheme, in contrast, can be used for ImageNet: see Appendix H.)

Here, we offer a different solution that makes use of random projections to efficiently approximate the Frobenius norm of the Jacobian.[3] This only introduces a constant time overhead and can be made very small in practice. When considering such an approximate algorithm, one naively must trade off efficiency against accuracy for computing the Jacobian, which ultimately trades computation time for robustness. Prior work by Varga et al. (2017) briefly considers an approach based on random projection, but without providing any analysis on the quality of the Jacobian approximation. Here, we describe our algorithm, analyze theoretical convergence guarantees, and verify empirically that there is only a negligible difference in model solution quality between training with the exact computation of the Jacobian as compared to training with the approximate algorithm, even when using a single random projection (see Figure 2).

Given that optimization is commonly gradient based, it is essential to efficiently compute gradients of the joint loss in Equation (5) and in particular of the squared Frobenius norm of the Jacobian. First, we note that automatic differentiation systems implement a function that computes the derivative of a vector such as $z$ with respect to any variables on which it depends, if the vector is first contracted with another fixed vector. To take advantage of this functionality, we rewrite the squared Frobienus norm as

$$||J(\boldsymbol{x})||_{\mathrm{F}}^2 = \mathrm{Tr}\left(JJ^{\mathrm{T}}\right) = \sum_{\{\boldsymbol{e}\}} \boldsymbol{e} J J^{\mathrm{T}} \boldsymbol{e}^{\mathrm{T}} = \sum_{\{\boldsymbol{e}\}} \left[\frac{\partial\left(\boldsymbol{e}\cdot\boldsymbol{z}\right)}{\partial\boldsymbol{x}}\right]^2, \qquad (6)$$

where a constant orthonormal basis, $\{\boldsymbol{e}\}$, of the $C$-dimensional output space was inserted in the second equality and the last equality follows from definition (2) and moving the constant vector inside the derivative. For each basis vector $\boldsymbol{e}$, the quantity in the last parenthesis can then be efficiently computed by differentiating the product, $\boldsymbol{e}\cdot\boldsymbol{z}$, with respect to input parameters, $\boldsymbol{x}$. Recycling that computational graph, the derivative of the squared Frobenius norm with respect to the model parameters, $\boldsymbol{\theta}$, can be computed through backpropagation with any use of automatic differentiation. Sokolić et al. (2017) essentially considers this exact computation, which requires backpropagating gradients through the model $C$ times to iterate over the $C$ orthonormal basis vectors $\{\boldsymbol{e}\}$. Ultimately, this incurs computational overhead that scales linearly with the output dimension $C$.

Instead, we further rewrite Equation (6) in terms of the expectation of an unbiased estimator

$$||J(\boldsymbol{x})||_{\mathrm{F}}^2 = C\,\mathbb{E}_{\hat{\boldsymbol{v}}\sim S^{C-1}}\left[||\hat{\boldsymbol{v}}\cdot J||^2\right], \qquad (7)$$

where the random vector $\hat{\boldsymbol{v}}$ is drawn from the $(C-1)$-dimensional unit sphere $S^{C-1}$. Using this relationship, we can use samples of $n_{\mathrm{proj}}$ random vectors $\hat{\boldsymbol{v}}^{\mu}$ to estimate the square of the norm as

$$||J(\boldsymbol{x})||_{\mathrm{F}}^2 \approx \frac{1}{n_{\mathrm{proj}}} \sum_{\mu=1}^{n_{\mathrm{proj}}} \left[\frac{\partial\left(\hat{\boldsymbol{v}}^{\mu}\cdot\boldsymbol{z}\right)}{\partial\boldsymbol{x}}\right]^2, \qquad (8)$$

which converges to the true value as $O(n_{\mathrm{proj}}^{-1/2})$. The derivation of Equation (7) and the calculation of its convergence make use of random-matrix techniques and are provided in Appendix B.

Finally, we expect that the fluctuations of our estimator can be suppressed by cancellations within a mini-batch. With nearly independent and identically distributed samples in a mini-batch of size

---

[3]In Appendix C, we give an alternative method for computing gradients of the Jacobian regularizer by using an analytically derived formula.

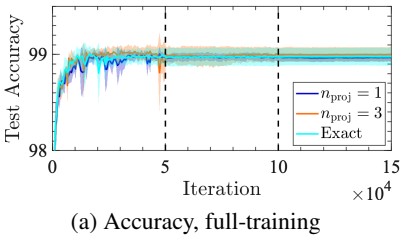 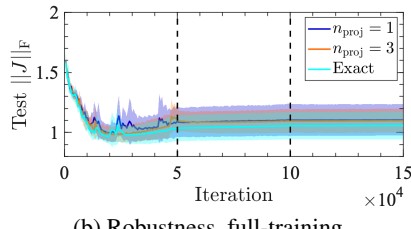

(a) Accuracy, full-training       (b) Robustness, full-training

Figure 2: **Comparison of Approximate to Exact Jacobian Regularizer.** The difference between the exact method (cyan) and the random projection method with $n_{\text{proj}} = 1$ (blue) and $n_{\text{proj}} = 3$ (red orange) is negligible both in terms of accuracy (a) and the norm of the input-output Jacobian (b) on the test set for LeNet' models trained on MNIST with $\lambda_{\text{JR}} = 0.01$. Shading indicates the standard deviation estimated over 5 distinct runs and dashed vertical lines signify the learning rate quenches.

---

**Algorithm 1** Efficient computation of the approximate gradient of the Jacobian regularizer.

---

**Inputs:** mini-batch of $|\mathcal{B}|$ examples $\boldsymbol{x}^\alpha$, model outputs $\boldsymbol{z}^\alpha$, and number of projections $n_{\text{proj}}$.
**Outputs:** Square of the Frobenius norm of the Jacobian $\mathcal{J}_F$ and its gradient $\nabla_{\boldsymbol{\theta}}\mathcal{J}_F$.
$\mathcal{J}_F = 0$
**for** $i = 1$ to $n_{\text{proj}}$ **do**
  $\{v_c^\alpha\} \sim \mathcal{N}(0, \mathbb{I})$  ▷ $(|\mathcal{B}|, C)$-dim tensor with each element sampled from a standard normal.
  $\hat{\boldsymbol{v}}^\alpha = \boldsymbol{v}^\alpha/||\boldsymbol{v}^\alpha||$           ▷ Uniform sampling from the unit sphere for each $\alpha$.
  $\boldsymbol{z}_{\text{flat}} = \text{Flatten}(\{\boldsymbol{z}^\alpha\}); \boldsymbol{v}_{\text{flat}} = \text{Flatten}(\{\hat{\boldsymbol{v}}^\alpha\})$       ▷ Flatten for parallelism.
  $Jv = \partial(\boldsymbol{z}_{\text{flat}} \cdot \boldsymbol{v}_{\text{flat}})/\partial \boldsymbol{x}^\alpha$
  $\mathcal{J}_F += C||Jv||^2/(n_{\text{proj}}|\mathcal{B}|)$
**end for**
$\nabla_{\boldsymbol{\theta}}\mathcal{J}_F = \partial\mathcal{J}_F/\partial\boldsymbol{\theta}$
**return** $\mathcal{J}_F, \nabla_{\boldsymbol{\theta}}\mathcal{J}_F$

---

$|\mathcal{B}| \gg 1$, we expect the error in our estimate to be of order $(n_{\text{proj}}|\mathcal{B}|)^{-1/2}$. In fact, as shown in Figure 2, with a mini-batch size of $|\mathcal{B}| = 100$, single projection yields model performance that is nearly identical to the exact method, with computational cost being reduced by orders of magnitude.

The complete algorithm is presented in Algorithm 1. With a straightforward implementation in PyTorch (Paszke et al., 2017) and $n_{\text{proj}} = 1$, we observed the computational cost of the training with the Jacobian regularization to be only $\approx 1.3$ times that of the standard SGD computation cost, while retaining all the practical benefits of the expensive exact method.[4]

## 3 EXPERIMENTS

In this section, we evaluate the effectiveness of Jacobian regularization on robustness. As all regularizers constrain the learning problem, we begin by confirming that our regularizer effectively reduces the value of the Frobenius norm of the Jacobian while simultaneously maintaining or improving generalization to an unseen test set. We then present our core result, that Jacobian regularization provides significant robustness against corruption of input data from both random and adversarial perturbations (Section 3.2). In the main text we present results mostly with the MNIST dataset; the corresponding experiments for the CIFAR-10 (Krizhevsky and Hinton, 2009) and ImageNet (Deng et al., 2009) datasets are relegated to Appendices E and H. The following specifications apply throughout our experiments:

**Datasets:** The MNIST data consist of black-white images of hand-written digits with 28-by-28 pixels, partitioned into 60,000 training and 10,000 test samples (LeCun et al., 1998). We preprocess the data by subtracting the mean (0.1307) and dividing by the variance (0.3081) of the training data.

---

[4]The costs are measured on a single NVIDIA GP100 for the LeNet' architecture on MNIST data. The computational efficiency depends on datasets and model architectures; the largest we have observed is a factor of $\approx 2$ increase in computational time for ResNet-18 on CIFAR-10 (Appendix E), which is still of order one.

Table 1: **Generalization on clean test data.** LeNet' models learned with varying amounts of training samples per class are eveluted on MNIST test set. Jacobian regularizer substantially reduces the norm of the Jacobian while retaining test accuracy. Errors indicate 95% confidence intervals over 5 distinct runs for full training and 15 for sub-sample training.

| | Test Accuracy ($\uparrow$) | | | | | $\|\|J\|\|_{\mathrm{F}}$ ($\downarrow$) |
| --- | --- | --- | --- | --- | --- | --- |
| | Samples per class | | | | | |
| Regularizer | 1 | 3 | 10 | 30 | All | All |
| No regularization | $49.2 \pm 1.9$ | $67.0 \pm 1.7$ | $83.3 \pm 0.7$ | $90.4 \pm 0.5$ | $98.9 \pm 0.1$ | $32.9 \pm 3.3$ |
| $L^2$ | $49.9 \pm 2.1$ | $68.1 \pm 1.9$ | $84.3 \pm 0.8$ | $91.2 \pm 0.5$ | $\mathbf{99.2 \pm 0.1}$ | $4.6 \pm 0.2$ |
| Dropout | $49.7 \pm 1.7$ | $67.4 \pm 1.7$ | $83.9 \pm 1.8$ | $91.6 \pm 0.5$ | $98.6 \pm 0.1$ | $21.5 \pm 2.3$ |
| Jacobian | $49.3 \pm 2.1$ | $68.2 \pm 1.9$ | $84.5 \pm 0.9$ | $91.3 \pm 0.4$ | $99.0 \pm 0.0$ | $\mathbf{1.1 \pm 0.1}$ |
| All Combined | $\mathbf{51.7 \pm 2.1}$ | $\mathbf{69.7 \pm 1.9}$ | $\mathbf{86.3 \pm 0.9}$ | $\mathbf{92.7 \pm 0.4}$ | $99.1 \pm 0.1$ | $1.2 \pm 0.0$ |

**Implementation Details:** For the MNIST dataset, we use the modernized version of LeNet-5 (LeCun et al., 1998), henceforth denoted LeNet' (see Appendix D for full details). We optimize using SGD with momentum, $\rho = 0.9$, and our supervised loss equals the standard cross-entropy with one-hot targets. The model parameters $\boldsymbol{\theta}$ are initialized at iteration $t = 0$ by the Xavier method (Glorot and Bengio, 2010) and the initial descent value is set to $\mathbf{0}$. The hyperparameters for all models are chosen to match reference implementations: the $L^2$ regularization coefficient (weight decay) is set to $\lambda_{\mathrm{WD}} = 5 \cdot 10^{-4}$ and the dropout rate is set to $p_{\mathrm{drop}} = 0.5$. The Jacobian regularization coefficient $\lambda_{\mathrm{JR}} = 0.01$, is chosen by optimizing for clean performance and robustness on the white noise perturbation. (See Appendix G for performance dependence on the coefficient $\lambda_{\mathrm{JR}}$.)

### 3.1 EVALUATING GENERALIZATION

The main goal of supervised learning involves generalizing from a training set to unseen test set. In dealing with such a distributional shift, overfitting to the training set and concomitant degradation in test performance is the central concern. For neural networks one of the most standard antidotes to this overfitting instability is $L^2$ regularization (Hinton, 1987; Krogh and Hertz, 1992; Zhang et al., 2018). More recently, dropout regularization has been proposed as another way to circumvent overfitting (Srivastava et al., 2014). Here we show how Jacobian regualarization can serve as yet another solution. This is also in line with the observed correlation between the input-output Jacobian and generalization performance (Novak et al., 2018).

**Generalizing within domain:** We first verify that in the clean case, where the test set is composed of unseen samples drawn from the same distribution as the training data, the Jacobian regularizer does not adversely affect classification accuracy. Table 1 reports performance on the MNIST test set for the LeNet' model trained on either a subsample or all of the MNIST train set, as indicated. When learning using all 60,000 training examples, the learning rate is initially set to $\eta_0 = 0.1$ with mini-batch size $|\mathcal{B}| = 100$ and then decayed ten-fold after each 50,000 SGD iterations; each simulation is run for 150,000 SGD iterations in total. When learning using a small subsample of the full training set, training is carried out using SGD with full batch and a constant learning rate $\eta = 0.01$, and the model performance is evaluated after 10,000 iterations. The main observation is that optimizing with the proposed Jacobian regularizer or the commonly used $L^2$ and dropout regularizers does not change performance on clean data within domain test samples in any statistically significant way. Notably, when few samples are available during learning, performance improved with increased regularization in the form of jointly optimizing over all criteria. Finally, in the right most column of Table 1, we confirm that the model trained with all data and regularized with the Jacobian minimization objective has an order of magnitude smaller Jacobian norm than models trained without Jacobian regularization. This indicates that while the model continues to make the same predictions on clean data, the margins around each prediction has increased as desired.

**Generalizing to a new domain:** We test the limits of the generalization provided by Jacobian regularization by evaluating an MNIST learned model on data drawn from a new target domain distribution – the USPS (Hull, 1994) test set. Here, models are trained on the MNIST data as above, and the USPS test dataset consists of 2007 black-white images of hand-written digits with

Table 2: **Generalization on clean test data from an unseen domain.** LeNet' models learned with all MNIST training data are evaluated for accuracy on data from the novel input domain of USPS test set. Here, each regularizer, including Jacobian, increases accuracy over an unregularized model. In addition, the regularizers may be combined for the strongest generalization effects. Averages and 95% confidence intervals are estimated over 5 distinct runs.

| No regularization | $L^2$ | Dropout | Jacobian | All Combined |
|---|---|---|---|---|
| $80.4 \pm 0.7$ | $83.3 \pm 0.8$ | $81.9 \pm 1.4$ | $81.3 \pm 0.9$ | $\mathbf{85.7 \pm 1.0}$ |

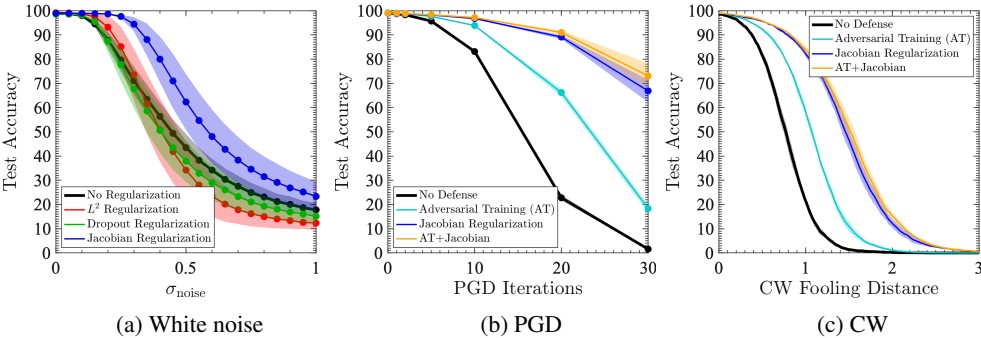

(a) White noise      (b) PGD      (c) CW

Figure 3: **Robustness against random and adversarial input perturbations.** This key result illustrates that Jacobian regularization significantly increases the robustness of a learned model with LeNet' architecture trained on the MNIST dataset. (a) Considering robustness under white noise perturbations, Jacobian minimization is the most effective regularizer. (b,c) Jacobian regularization alone outperforms an adversarial training defense (base models all include $L^2$ and dropout regularization). Shades indicate standard deviations estimated over 5 distinct runs.

16-by-16 pixels; images are upsampled to 28-by-28 pixels using bilinear interpolation and then preprocessed following the MNIST protocol stipulated above. Table 2 offers preliminary evidence that regularization, of each of the three forms studied, can be used to learn a source model which better generalizes to an unseen target domain. We again find that the regularizers may be combined to increase the generalization property of the model. Such a regularization technique can be immediately combined with state-of-the-art domain adaptation techniques to achieve further gains.

## 3.2 EVALUATING UNDER DATA CORRUPTION

This section showcases the main robustness results of the Jacobian regularizer, highlighted in the case of both random and adversarial input perturbations.

**Random Noise Corruption:** The real world can differ from idealized experimental setups and input data can become corrupted by various natural causes such as random noise and occlusion. Robust models should minimize the impact of such corruption. As one evaluation of stability to natural corruption, we perturb each test input image $\boldsymbol{x}$ to $\widetilde{\boldsymbol{x}} = \lceil \boldsymbol{x} + \boldsymbol{\epsilon} \rceil_{\mathrm{crop}}$ where each component of the perturbation vector is drawn from the normal distribution with variance $\sigma_{\mathrm{noise}}$ as

$$\epsilon_i \sim \mathcal{N}(0, \sigma_{\mathrm{noise}}^2), \tag{9}$$

and the perturbed image is then clipped to fit into the range $[0, 1]$ before preprocessing. As in the domain-adaptation experiment above, models are trained on the clean MNIST training data and then tested on corrupted test data. Results in Figure 3a show that models trained with the Jacobian regularization is more robust against white noise than others. This is in line with – and indeed quantitatively validates – the embiggening of decision cells as shown in Figure 1.

**Adversarial Perturbations:** The world is not only imperfect but also possibly filled with evil agents that can deliberately attack models. Such adversaries seek a small perturbation to each input example that changes the model predictions while also being imperceptible to humans. Obtaining the actual smallest perturbation is likely computationally intractable, but there exist many tractable approxima-

tions. The simplest attack is the white-box untargeted fast gradient sign method (FGSM) (Goodfellow et al., 2014), which distorts the image as $\widetilde{\boldsymbol{x}} = \lceil \boldsymbol{x} + \boldsymbol{\epsilon} \rceil_{\mathrm{crop}}$ with

$$\epsilon_i = \varepsilon_{\mathrm{FGSM}} \cdot \mathrm{sign}\left( \sum_c \frac{\partial \mathcal{L}_{\mathrm{super}}}{\partial z_c} J_{c;i} \right). \tag{10}$$

This attack aggregates nonzero components of the input-output Jacobian to a substantial effect by adding them up with a consistent sign. In Figure 3b we consider a stronger attack, projected gradient descent (PGD) method (Kurakin et al., 2016; Madry et al., 2017), which iterates the FGSM attack in Equation (10) $k$ times with fixed amplitude $\varepsilon_{\mathrm{FGSM}} = 1/255$ while also requiring each pixel value to be within $32/255$ away from the original value. Even stronger is the Carlini-Wagner (CW) attack (Carlini and Wagner, 2017) presented in Figure 3c, which yields more reliable estimates of distance to the closest decision boundary (see Appendix F). Results unequivocally show that models trained with the Jacobian regularization is again more resilient than others. As a baseline defense benchmark, we implemented adversarial training, where the training image is corrupted through the FGSM attack with uniformly drawn amplitude $\varepsilon_{\mathrm{FGSM}} \in [0, 0.01]$; the Jacobian regularization can be combined with this defense mechanism to further improve the robustness.[5] Appendix A additionally depicts decision cells in adversarial directions, further illustrating the stabilizing effect of the Jacobian regularizer.

## 4 RELATED WORK

To our knowledge, double backpropagation (Drucker and LeCun, 1991; 1992) is the earliest attempt to penalize large derivatives with respect to input data, in which $(\partial \mathcal{L}_{\mathrm{super}}/\partial \boldsymbol{x})^2$ is added to the loss in order to reduce the generalization gap.[6] Different incarnations of a similar idea have appeared in the following decades (Simard et al., 1992; Mitchell and Thrun, 1993; Aires et al., 1999; Rifai et al., 2011; Gulrajani et al., 2017; Yoshida and Miyato, 2017; Czarnecki et al., 2017; Jakubovitz and Giryes, 2018). Among them, Jacobian regularization as formulated herein was proposed by Gu and Rigazio (2014) to combat against adversarial attacks. However, the authors did not implement it due to a computational concern – resolved by us in Section 2 – and instead layer-wise Jacobians were penalized. Unfortunately, minimizing layer-wise Jacobians puts a stronger constraint on model capacity than minimizing the input-output Jacobian. In fact, several authors subsequently claimed that the layer-wise regularization degrades test performance on clean data (Goodfellow et al., 2014; Papernot et al., 2016b) and results in marginal improvement of robustness (Carlini and Wagner, 2017).

Very recently, full Jacobian regularization was implemented in Sokolić et al. (2017), but in an inefficient manner whose computational overhead for computing gradients scales linearly with the number of output classes $C$ compared to unregularized optimization, and thus they had to resort back to the layer-wise approximation above for the task with a large number of output classes. This computational problem was resolved by Varga et al. (2017) in exactly the same way as our approach (referred to as spherical SpectReg in Varga et al. (2017)). As emphasized in Section 2, we performed more thorough theoretical and empirical convergence analysis and showed that there is practically no difference in model solution quality between the exact and random projection method in terms of test accuracy and stability. Further, both of these two references deal only with the generalization property and did not fully explore strong distributional shifts and noise/adversarial defense. In particular, we have visualized (Figure 1) and quantitatively borne out (Section 3) the stabilizing effect of Jacobian regularization on classification margins of a nonlinear neural network.

---

[5]We also tried the defensive distillation technique of Papernot et al. (2016b). While the model trained with distillation temperature $T = 100$ and attacked with $T = 1$ appeared robust against FGSM/PGD adversaries, it was fragile once attacked at $T = 100$ and thus cannot be robust against white-box attacks. This is in line with the numerical precision issue observed by Carlini and Wagner (2016).

[6]This approach was slightly generalized in Lyu et al. (2015) in the context of adversarial defense; see also Ororbia II et al. (2016); Ross and Doshi-Velez (2018).

# 5 CONCLUSION

In this paper, we motivated Jacobian regularization as a task-agnostic method to improve stability of models against perturbations to input data. Our method is simply implementable in any open source automatic differentiation system, and additionally we have carefully shown that the approximate nature of the random projection is virtually negligible. Furthermore, we have shown that Jacobian regularization enlarges the size of decision cells and is practically effective in improving the generalization property and robustness of the models, which is especially useful for defense against input-data corruption. We hope practitioners will combine our Jacobian regularization scheme with the arsenal of other tricks in machine learning and prove it useful in pushing the (decision) boundary of the field and ensuring stable deployment of models in everyday life.

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

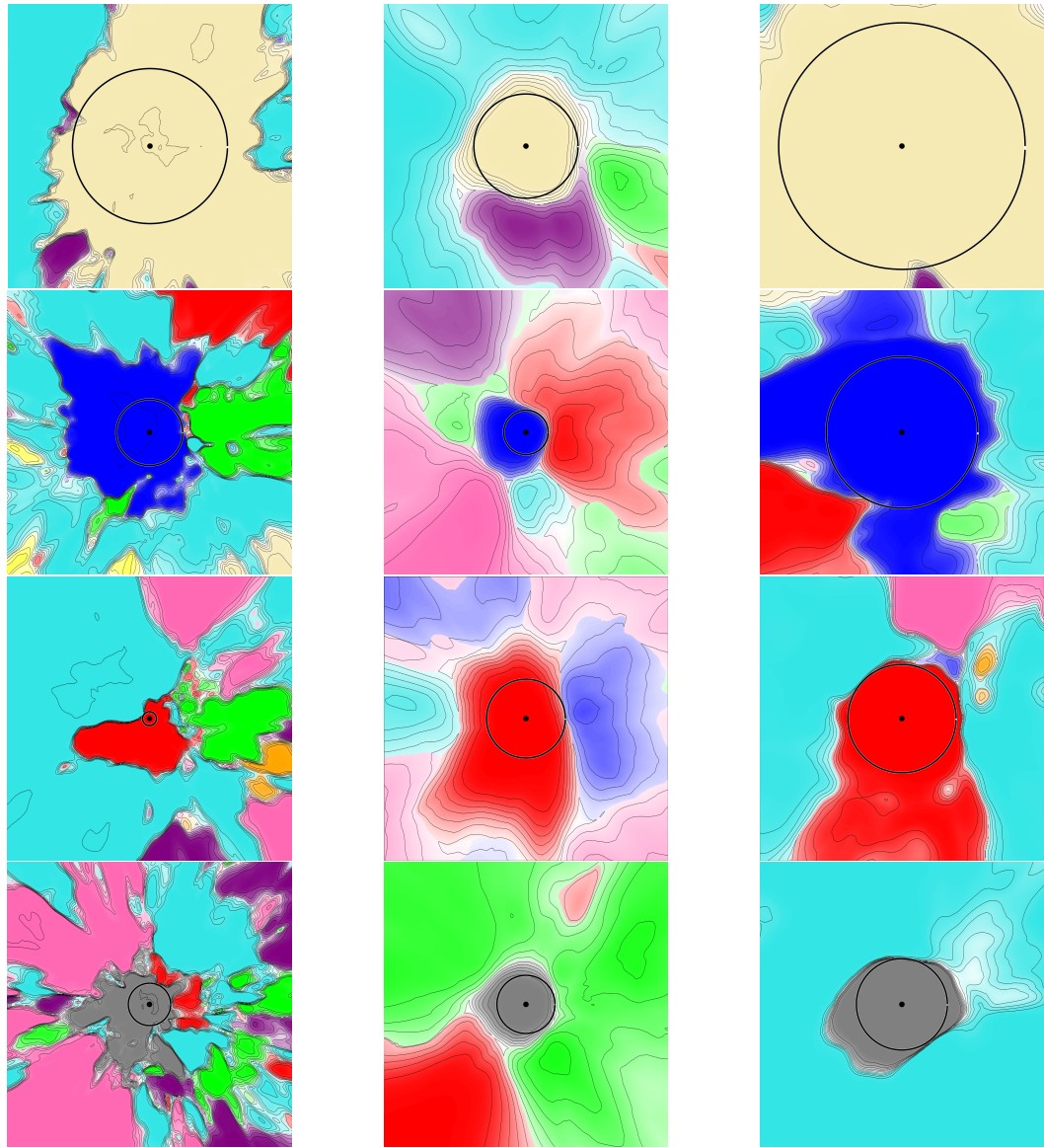

Figure S1: **Cross sections of decision cells in the input space for LeNet' models trained on the MNIST dataset along random hyperplanes.** Figure specifications are same as in Figure 1. (Left) No regularization. (Middle) $L^2$ regularization with $\lambda_{\mathrm{WD}} = 0.0005$ . (Right) Jacobian regularization with $\lambda_{\mathrm{JR}} = 0.01$.

## A  GALLERY OF DECISION CELLS

We show in Figure S1 plots similar to the ones shown in Figure 1 in the main text, but with different seeds for training models and around different test data points. Additionally, shown in Figure S2 are similar plots but with different scheme for hyperplane slicing, based on adversarial directions. Interestingly, the adversarial examples constructed with unprotected model do not fool the model trained with Jacobian regularization.

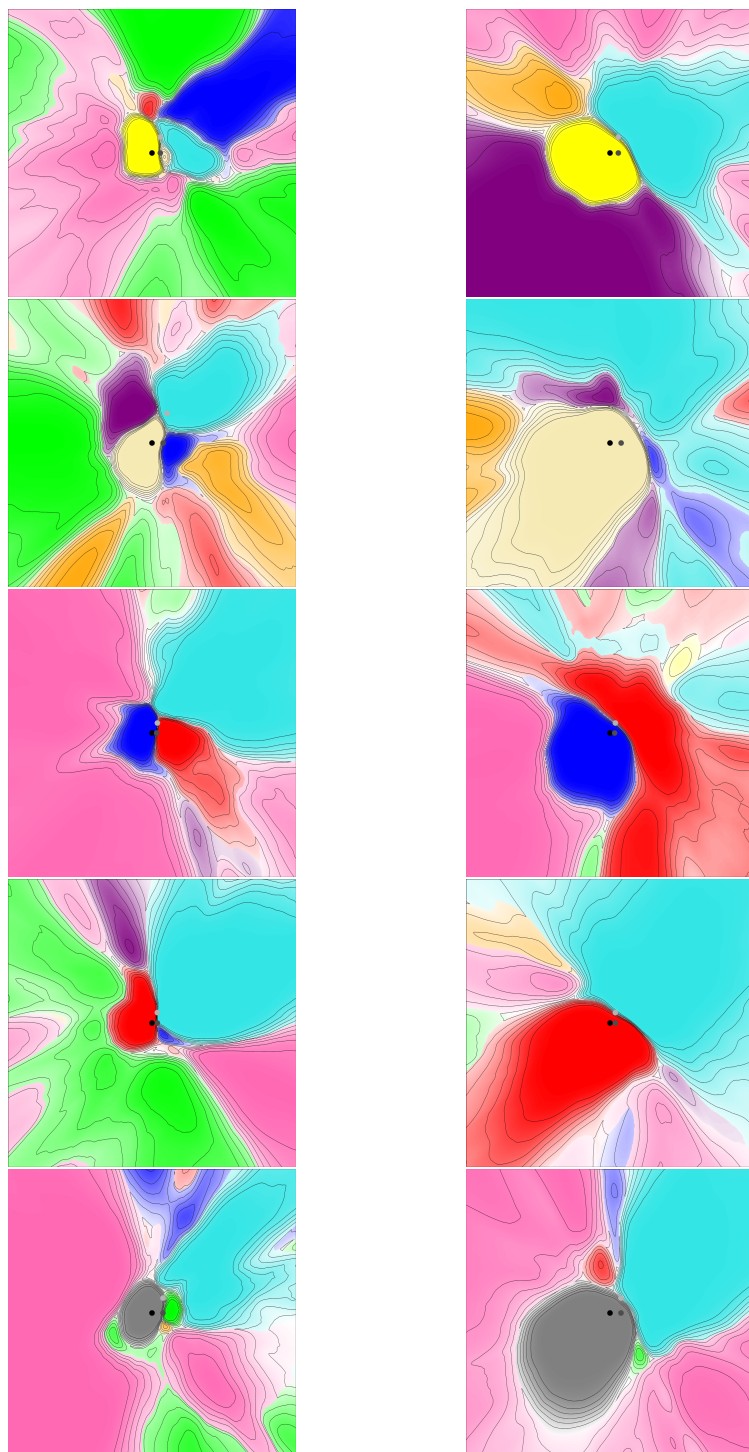

Figure S2: **Cross sections of decision cells in the input space for LeNet' models trained on the MNIST dataset along adversarial hyperplanes.** Namely, given a test sample (black dot), the hyperplane through it is spanned by two adversarial examples identified through FGSM, one for the model trained with $L^2$ regularization $\lambda_{\mathrm{WD}} = 0.0005$ and dropout rate $0.5$ but no defense (dark-grey dot; left figure) and the other for the model with the same standard regularization methods plus Jacobian regularization $\lambda_{\mathrm{JR}} = 0.01$ and adversarial training (white-grey dot; right figure).

## B  ADDITIONAL DETAILS FOR EFFICIENT ALGORITHM

Let us denote by $\mathbb{E}_{\hat{\boldsymbol{v}} \sim S^{C-1}} [F(\hat{\boldsymbol{v}})]$ the average of the arbitrary function $F$ over $C$-dimensional vectors $\hat{\boldsymbol{v}}$ sampled uniformly from the unit sphere $S^{C-1}$. As in Algorithm 1, such a unit vector can be sampled by first sampling each component $v_c$ from the standard normal distribution $\mathcal{N}(0,1)$ and then normalizing it as $\hat{\boldsymbol{v}} \equiv \boldsymbol{v}/||\boldsymbol{v}||$. In our derivation, the following formula proves useful:

$$\mathbb{E}_{\hat{\boldsymbol{v}} \sim S^{C-1}} [F(\hat{\boldsymbol{v}})] = \int d\mu(O) F(O\boldsymbol{e}), \tag{11}$$

where $\boldsymbol{e}$ is an arbitrary $C$-dimensional unit vector and $\int d\mu(O) [\ldots]$ is an integral over orthogonal matrices $O$ over the Haar measure with normalization $\int d\mu(O) [1] = 1$.

First, let us derive Equation (7). Using Equation (11), the square of the Frobenius norm can then be written as

$$
\begin{aligned}
||J(\boldsymbol{x})||_{\mathrm{F}}^2 &= \operatorname{Tr}(JJ^{\mathrm{T}}), \\
&= \int d\mu(O) \operatorname{Tr}(OJJ^{\mathrm{T}}O^{\mathrm{T}}), \\
&= \int d\mu(O) \sum_{\{\boldsymbol{e}\}} \boldsymbol{e}OJJ^{\mathrm{T}}O^{\mathrm{T}}\boldsymbol{e}^{\mathrm{T}}, \\
&= \sum_{\{\boldsymbol{e}\}} \mathbb{E}_{\hat{\boldsymbol{v}} \sim S^{C-1}} [\hat{\boldsymbol{v}}JJ^{\mathrm{T}}\hat{\boldsymbol{v}}^{\mathrm{T}}], \\
&= C\,\mathbb{E}_{\hat{\boldsymbol{v}} \sim S^{C-1}} [\hat{\boldsymbol{v}}JJ^{\mathrm{T}}\hat{\boldsymbol{v}}^{\mathrm{T}}], 
\end{aligned} \tag{12}
$$

where in the second line we insert the identity matrix in the form $\mathbb{I} = O^T O$ and make use of the cyclicity of the trace; in the third line we rewrite the trace as a sum over an orthonormal basis $\{\boldsymbol{e}\}$ of the $C$-dimensional output space; in the forth line Equation (11) was used; and in the last line we note that the expectation no longer depends on the basis vectors $\boldsymbol{e}$ and perform the trivial sum. This completes the derivation of Equation (7).

Next, let us compute the variance of our estimator. Using tricks as before, but in reverse order, yields

$$\operatorname{var}(C\,\hat{\boldsymbol{v}}JJ^{\mathrm{T}}\hat{\boldsymbol{v}}^{\mathrm{T}}) \equiv C^2\,\mathbb{E}_{\hat{\boldsymbol{v}} \sim S^{C-1}} \left[(\hat{\boldsymbol{v}}JJ^{\mathrm{T}}\hat{\boldsymbol{v}}^{\mathrm{T}})^2\right] - ||J(\boldsymbol{x})||_{\mathrm{F}}^4, \tag{13}$$

$$= C^2 \int d\mu(O) \left[\boldsymbol{e}OJJ^{\mathrm{T}}O^{\mathrm{T}}\boldsymbol{e}^{\mathrm{T}}\boldsymbol{e}OJJ^{\mathrm{T}}O^{\mathrm{T}}\boldsymbol{e}^{\mathrm{T}}\right] - ||J(\boldsymbol{x})||_{\mathrm{F}}^4.$$

In this form, we use the following formula (Collins and Śniady, 2006; Collins and Matsumoto, 2009) to evaluate the first term[7]

$$\int d\mu(O)\, O_{c_1 c_5} O_{c_6 c_2}^{T} O_{c_3 c_7} O_{c_8 c_4}^{T} = \tag{14}$$

$$\frac{C+1}{C(C-1)(C+2)} \left(\delta_{c_1 c_2}\delta_{c_3 c_4}\delta_{c_5 c_6}\delta_{c_7 c_8} + \delta_{c_1 c_3}\delta_{c_2 c_4}\delta_{c_5 c_7}\delta_{c_6 c_8} + \delta_{c_1 c_4}\delta_{c_2 c_3}\delta_{c_5 c_8}\delta_{c_6 c_7}\right)$$

$$-\frac{1}{C(C-1)(C+2)} \left(\delta_{c_1 c_2}\delta_{c_3 c_4}\delta_{c_5 c_7}\delta_{c_6 c_8} + \delta_{c_1 c_2}\delta_{c_3 c_4}\delta_{c_5 c_8}\delta_{c_6 c_7} + \delta_{c_1 c_3}\delta_{c_2 c_4}\delta_{c_5 c_6}\delta_{c_7 c_8}\right.$$

$$\left. +\delta_{c_1 c_3}\delta_{c_2 c_4}\delta_{c_5 c_8}\delta_{c_6 c_7} + \delta_{c_1 c_4}\delta_{c_2 c_3}\delta_{c_5 c_6}\delta_{c_7 c_8} + \delta_{c_1 c_4}\delta_{c_2 c_3}\delta_{c_5 c_7}\delta_{c_6 c_8}\right).$$

After the dust settles with various cancellations, the expression for the variance simplifies to

$$\operatorname{var}(C\,\hat{\boldsymbol{v}}JJ^{\mathrm{T}}\hat{\boldsymbol{v}}^{\mathrm{T}}) = \frac{2\,C}{(C+2)} \operatorname{Tr}(JJ^{\mathrm{T}}JJ^{\mathrm{T}}) - \frac{2}{(C+2)}||J(\boldsymbol{x})||_{\mathrm{F}}^4. \tag{15}$$

We can strengthen our claim by using the relation $||AB||_{\mathrm{F}}^2 \leq ||A||_{\mathrm{F}}^2 ||B||_{\mathrm{F}}^2$ with $A = J$ and $B = J^{\mathrm{T}}$, which yields $\operatorname{Tr}(JJ^{\mathrm{T}}JJ^{\mathrm{T}}) \leq ||J(\boldsymbol{x})||_{\mathrm{F}}^4$ and in turn bounds the variance divided by the square of the mean as

$$\frac{\operatorname{var}(C\,\hat{\boldsymbol{v}}JJ^{\mathrm{T}}\hat{\boldsymbol{v}}^{\mathrm{T}})}{[\operatorname{mean}(C\,\hat{\boldsymbol{v}}JJ^{\mathrm{T}}\hat{\boldsymbol{v}}^{\mathrm{T}})]^2} \leq 2\left(\frac{C-1}{C+2}\right). \tag{16}$$

---

[7]We thank Nick Hunter-Jones for providing us with the inelegant but concretely actionable form of this integral.

The right-hand side is independent of $J$ and thus independent of the details of model architecture and particular data set considered.

In the end, the relative error of the random-projection estimate for $||J(\boldsymbol{x})||_{\mathrm{F}}^2$ with $n_{\mathrm{proj}}$ random vectors will diminish as some order-one number divided by $n_{\mathrm{proj}}^{-1/2}$. In addition, upon averaging $||J(\boldsymbol{x})||_{\mathrm{F}}^2$ over a mini-batch of samples of size $|\mathcal{B}|$, we expect the relative error of the Jacobian regularization term to be additionally suppressed by $\sim 1/\sqrt{|\mathcal{B}|}$.

Finally, we speculate that in the large-$C$ limit – possibly relevant for large-class datasets such as the ImageNet (Deng et al., 2009) – there might be additional structure in the Jacobian traces (e.g. the central-limit concentration) that leads to further suppression of the variance.

## C   Cyclopropagation for Jacobian Regularization

It is also possible to derive a closed-form expression for the derivative of the Jacobian regularizer, thus bypassing any need for random projections while maintaining computational efficiency. The expression is here derived for multilayer perceptron, though we expect similar computations may be done for other models of interest. We provide full details in case one may find it practically useful to implement explicitly in any open-source packages or generalize it to other models.

Let us denote the input $x_i$ and the output $z_c = z_c^{(L)}$ where (identifying $\{i\} = \{i_0\} = \{1, \ldots, I\}$ and $\{c\} = \{i_L\} = \{1, \ldots, C\}$)

$$z_{i_0}^{(0)} \equiv x_{i_0}, \tag{17}$$

$$\hat{z}_{i_\ell}^{(\ell)} = \left( \sum_{i_{\ell-1}} w_{i_\ell, i_{\ell-1}}^{(\ell)} z_{i_{\ell-1}}^{(\ell-1)} \right) + b_{i_\ell}^{(\ell)} \quad \text{for} \quad \ell = 1, \ldots, L \tag{18}$$

$$z_{i_\ell}^{(\ell)} = \sigma\left(\hat{z}_{i_\ell}^{(\ell)}\right) \quad \text{for} \quad \ell = 1, \ldots, L. \tag{19}$$

Defining the layer-wise Jacobian as

$$J_{i_\ell, i_{\ell-1}}^{(\ell)} \equiv \frac{\partial z_{i_\ell}^{(\ell)}}{\partial z_{i_{\ell-1}}^{(\ell-1)}} = \sigma'\left(\hat{z}_{i_\ell}^{(\ell)}\right) w_{i_\ell, i_{\ell-1}}^{(\ell)} \quad \text{(no summation)}, \tag{20}$$

the total input-output Jacobian is given by

$$J_{i_L, i_0} \equiv \frac{\partial z_{i_L}^{(L)}}{\partial z_{i_0}} = \left[ J^{(L)} J^{(L-1)} \cdots J^{(1)} \right]_{i_L, i_0}. \tag{21}$$

The Jacobian regularizer of interest is defined as (up to the magnitude coefficient $\lambda_{\mathrm{JR}}$)

$$\mathcal{R}_{\mathrm{JR}} \equiv \frac{1}{2} ||J||_{\mathrm{F}}^2 \equiv \frac{1}{2} \sum_{i_0, i_L} (J_{i_L, i_0})^2 = \frac{1}{2} \mathrm{Tr}\left[J^{\mathrm{T}} J\right]. \tag{22}$$

Its derivatives with respect to biases and weights are denoted as

$$\widetilde{B}_{j_\ell}^{(\ell)} \equiv \frac{\partial \mathcal{R}_{\mathrm{JR}}}{\partial b_{j_\ell}^{(\ell)}}, \tag{23}$$

$$\widetilde{W}_{j_\ell, j_{\ell-1}}^{(\ell)} \equiv \frac{\partial \mathcal{R}_{\mathrm{JR}}}{\partial w_{j_\ell, j_{\ell-1}}^{(\ell)}}. \tag{24}$$

Some straightforward algebra then yields

$$\widetilde{B}_{j_\ell}^{(\ell)} = \left[ \frac{\widetilde{B}^{(\ell+1)}}{\sigma'(\hat{z}^{(\ell+1)})} J^{(\ell+1)} \right]_{j_\ell} \sigma'(\hat{z}_{j_\ell}^{(\ell)}) + \frac{\sigma''\left(\hat{z}_{j_\ell}^{(\ell)}\right)}{\sigma'\left(\hat{z}_{j_\ell}^{(\ell)}\right)} \left[ J^{(\ell)} \cdots J^{(1)} \cdot J^{\mathrm{T}} \cdot J^{(L)} \cdots J^{(\ell+1)} \right]_{j_\ell, j_\ell}, \tag{25}$$

and

$$\widetilde{W}^{(\ell)}_{j_\ell, j_{\ell-1}} = \widetilde{B}^{(\ell)}_{j_\ell} z^{(\ell-1)}_{j_{\ell-1}} + \sigma'\left(\hat{z}^{(\ell)}_{j_\ell}\right) \left[ J^{(\ell-1)} \cdots J^{(1)} \cdot J^{\mathrm{T}} \cdot J^{(L)} \cdots J^{(\ell+1)} \right]_{j_{\ell-1}, j_\ell}, \quad (26)$$

where we have set

$$\widetilde{B}^{(L+1)}_{j_{L+1}} = J^{(L+1)}_{j_{L+1}} = 0. \quad (27)$$

Algorithmically, we can iterate the following steps for $\ell = L, L-1, \ldots, 1$:

1. Compute[8]

$$\Omega^{(\ell)}_{j_{\ell-1}, j_\ell} \equiv \left[ J^{(\ell-1)} \cdots J^{(1)} \cdot J^{\mathrm{T}} \cdot J^{(L)} \cdots J^{(\ell+1)} \right]_{j_{\ell-1}, j_\ell}. \quad (28)$$

2. Compute

$$\frac{\partial R}{\partial b^{(\ell)}_{j_\ell}} = \widetilde{B}^{(\ell)}_{j_\ell} = \left[ \frac{\widetilde{B}^{(\ell+1)}}{\sigma'(\hat{z}^{(\ell+1)})} J^{(\ell+1)} \right]_{j_\ell} \sigma'(\hat{z}^{(\ell)}_{j_\ell}) + \sigma''\left(\hat{z}^{(\ell)}_{j_\ell}\right) \sum_{j_{\ell-1}} w^{(\ell)}_{j_\ell, j_{\ell-1}} \Omega^{(\ell)}_{j_{\ell-1}, j_\ell}. \quad (29)$$

3. Compute

$$\frac{\partial R}{\partial w^{(\ell)}_{j_\ell, j_{\ell-1}}} = \widetilde{W}^{(\ell)}_{j_\ell, j_{\ell-1}} = \widetilde{B}^{(\ell)}_{j_\ell} z^{(\ell-1)}_{j_{\ell-1}} + \sigma'\left(\hat{z}^{(\ell)}_{j_\ell}\right) \Omega^{(\ell)}_{j_{\ell-1}, j_\ell}. \quad (30)$$

Note that the layer-wise Jacobians, $J^{(\ell)}$'s, are calculated within the standard backpropagation algorithm. The core of the algorithm is in the computation of $\Omega^{(\ell)}_{j_{\ell-1}, j_\ell}$ in Equation (28). It is obtained by first backpropagating from $\ell - 1$ to 1, then forwardpropagating from 1 to $L$, and finally backpropagating from $L$ to $\ell + 1$. It thus makes the cycle around $\ell$, hence the name cyclopropagation.

## D  DETAILS FOR MODEL ARCHITECTURES

In order to describe architectures of our convolutional neural networks in detail, let us associate a tuple $[F, C_{\mathrm{in}} \to C_{\mathrm{out}}, S, P; M]$ to a convolutional layer with filter width $F$, number of in-channels $C_{\mathrm{in}}$ and out-channels $C_{\mathrm{out}}$, stride $S$, and padding $P$, followed by nonlinear activations and then a max-pooling layer of width $M$ (note that $M = 1$ corresponds to no pooling). Let us also associate a pair $[N_{\mathrm{in}} \to N_{\mathrm{out}}]$ to a fully-connected layer passing $N_{\mathrm{in}}$ inputs into $N_{\mathrm{out}}$ units with activations and possibly dropout.

With these notations, our LeNet' model used for the MNIST experiments consists of a $(28, 28, 1)$ input followed by a convolutional layer with $[5, 1 \to 6, 1, 2; 2]$, another one with $[5, 6 \to 16, 1, 0; 2]$, a fully-connected layer with $[2100 \to 120]$ and dropout rate $p_{\mathrm{drop}}$, another fully-connected layer with $[120 \to 84]$ and dropout rate $p_{\mathrm{drop}}$, and finally a fully-connected layer with $[84 \to 10]$, yielding 10-dimensional output logits. For our nonlinear activations, we use the hyperbolic tangent.

For the CIFAR-10 dataset, we use the model architecture specified in the paper on defensive distillation (Papernot et al., 2016b), abbreviated as DDNet. Specifically, the model consists of a $(32, 32, 3)$ input followed by convolutional layers with $[3, 3 \to 64, 1, 0; 1]$, $[3, 64 \to 64, 1, 0; 2]$, $[3, 64 \to 128, 1, 0; 1]$, and $[3, 128 \to 128, 1, 0; 2]$, and then fully-connected layers with $[3200 \to 256]$ and dropout rate $p_{\mathrm{drop}}$, with $[256 \to 256]$ and dropout rate $p_{\mathrm{drop}}$, and with $[256 \to 10]$, again yielding 10-dimensional output logits. All activations are rectified linear units.

In addition, we experiment with a version of ResNet-18 (He et al., 2016) modified for the 32-by-32 input size of CIFAR-10 and shown to achieve strong performance on clean image recognition.[9] For this architecture, we use the standard PyTorch initialization of the parameters. Data preprocessing and optimization hyperparameters for both architectures are specified in the next section.

For our ImageNet experiments, we use the standard ResNet-18 model available within PyTorch (torchvision.models.resnet) together with standard weight initialization.

Note that there is typically no dropout regularization in the ResNet models but we still examine the effect of $L^2$ regularization in addition to Jacobian regularization.

---

[8]For $\ell = 1$, the part $J^{(\ell-1)} \cdots J^{(1)}$ is vacuous. Similarly, for $\ell = L$, the part $J^{(L)} \cdots J^{(\ell+1)}$ is vacuous.

[9]Model available at: https://github.com/kuangliu/pytorch-cifar.

Table 3: **Generalization on clean test data.** DDNet models learned with varying amounts of training samples per class are evaluated on CIFAR-10 test set. Jacobian regularizer substantially reduces the norm of the Jacobian. Errors indicate 95% confidence intervals over 5 distinct runs for full training and 15 for sub-sample training.

| | Test Accuracy ($\uparrow$) | | | | | $||J||_{\mathrm{F}}$ ($\downarrow$) |
|---|---|---|---|---|---|---|
| | Samples per class | | | | | |
| Regularizer | 1 | 3 | 10 | 30 | All | All |
| No regularization | $12.9 \pm 0.7$ | $15.5 \pm 0.7$ | $20.5 \pm 1.3$ | $26.6 \pm 1.0$ | $76.8 \pm 0.4$ | $115.1 \pm 1.8$ |
| $L^2$ | $13.9 \pm 1.0$ | $14.6 \pm 1.1$ | $20.5 \pm 1.0$ | $26.6 \pm 1.2$ | $77.8 \pm 0.2$ | $29.4 \pm 0.5$ |
| Dropout | $12.9 \pm 1.4$ | $17.8 \pm 0.6$ | $24.4 \pm 1.0$ | $31.4 \pm 0.5$ | $\mathbf{80.7 \pm 0.4}$ | $184.2 \pm 4.8$ |
| Jacobian | $14.9 \pm 1.0$ | $18.3 \pm 1.0$ | $23.7 \pm 0.8$ | $30.0 \pm 0.6$ | $75.4 \pm 0.2$ | $\mathbf{4.0 \pm 0.0}$ |
| All Combined | $\mathbf{15.0 \pm 1.1}$ | $\mathbf{19.6 \pm 0.9}$ | $\mathbf{26.1 \pm 0.6}$ | $\mathbf{33.4 \pm 0.6}$ | $78.6 \pm 0.2$ | $5.2 \pm 0.0$ |

# E    RESULTS FOR CIFAR-10

Following specifications apply throughout this section for CIFAR-10 experiments with DDNet and ResNet-18 model architectures (see Appendix D).

- Datasets: the CIFAR-10 dataset consists of color images of objects – divided into ten categories – with 32-by-32 pixels in each of 3 color channels, each pixel ranging in $[0, 1]$, partitioned into 50,000 training and 10,000 test samples (Krizhevsky and Hinton, 2009). The images are preprocessed by uniformly subtracting 0.5 and multiplying by 2 so that each pixel ranges in $[-1, 1]$.

- Optimization: essentially same as for the LeNet' on MNIST, except the initial learning rate for full training. Namely, model parameters $\boldsymbol{\theta}$ are initialized at iteration $t = 0$ by the Xavier method (Glorot and Bengio, 2010) for DDNet and standard PyTorch initialization for ResNet-18, along with the zero initial velocity $\mathbf{v}(t = 0) = \mathbf{0}$. They evolve under the SGD dynamics with momentum $\rho = 0.9$, and for the supervised loss we use cross-entropy with one-hot targets. For training with the full training set, mini-batch size is set as $|\mathcal{B}| = 100$, and the learning rate $\eta$ is initially set to $\eta_0 = 0.01$ for the DDNet and $\eta_0 = 0.1$ for the ResNet-18 and in both cases quenched ten-fold after each 50,000 SGD iterations; each simulation is run for 150,000 SGD iterations in total. For few-shot learning, training is carried out using full-batch SGD with a constant learning rate $\eta = 0.01$, and model performance is evaluated after 10,000 iterations.

- Hyperparameters: the same values are inherited from the experiments for LeNet' on the MNIST and no tuning was performed. Namely, the weight decay coefficient $\lambda_{\mathrm{WD}} = 5 \cdot 10^{-4}$; the dropout rate $p_{\mathrm{drop}} = 0.5$; the Jacobian regularization coefficient $\lambda_{\mathrm{JR}} = 0.01$; and adversarial training with uniformly drawn FGSM amplitude $\varepsilon_{\mathrm{FGSM}} \in [0, 0.01]$.

The results relevant for generalization properties are shown in Table S3. One difference from the MNIST counterparts in the main text is that dropout improves test accuracy more than $L^2$ regularization. Meanwhile, for both setups the order of stability measured by $||J||_{\mathrm{F}}$ on the test set more or less stays the same. Most importantly, turning on the Jacobian regularizer improves the stability by orders of magnitude, and combining it with other regularizers do not compromise this effect.

The results relevant for robustness against input-data corruption are plotted in Figures S3 and S4. The success of the Jacobian regularizer is retained for the white-noise and CW adversarial attack. For the PGD attack results are mixed at high degradation level when Jacobian regularization is combined with adversarial training. This might be an artifact stemming from the simplicity of the PGD search algorithm, which overestimates the shortest distance to adversarial examples in comparison to the CW attack (see Appendix F), combined with Jacobian regularization's effect on simplifying the loss landscape with respect to the input space that the attack methods explore.

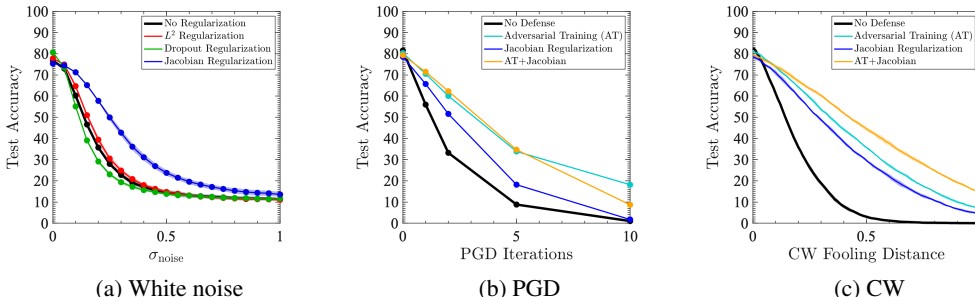

(a) White noise        (b) PGD        (c) CW

Figure S3: **Robustness against random and adversarial input perturbations for DDNet models trained on the CIFAR-10 dataset.** Shades indicate standard deviations estimated over 5 distinct runs. (a) Comparison of regularization methods for robustness to white noise perturbations. (b,c) Comparison of different defense methods against adversarial attacks (all models here equipped with $L^2$ and dropout regularization).

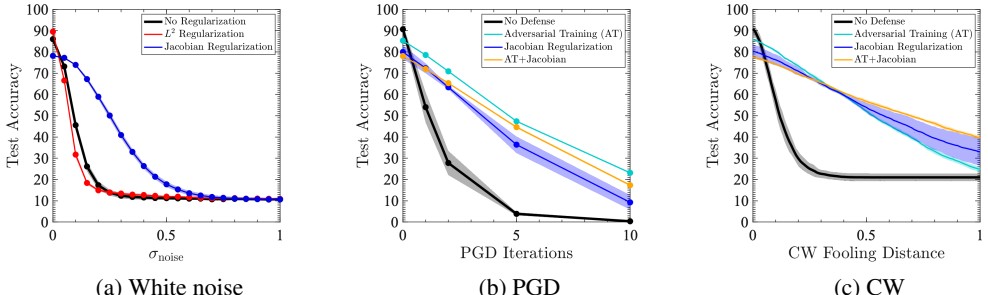

(a) White noise        (b) PGD        (c) CW

Figure S4: **Robustness against random and adversarial input perturbations for ResNet-18 models trained on the CIFAR-10 dataset.** Shades indicate standard deviations estimated over 5 distinct runs. (a) Comparison of regularization methods for robustness to white noise perturbations. (b,c) Comparison of different defense methods against adversarial attacks (all models here equipped with $L^2$ regularization but not dropout: see Appendix D).

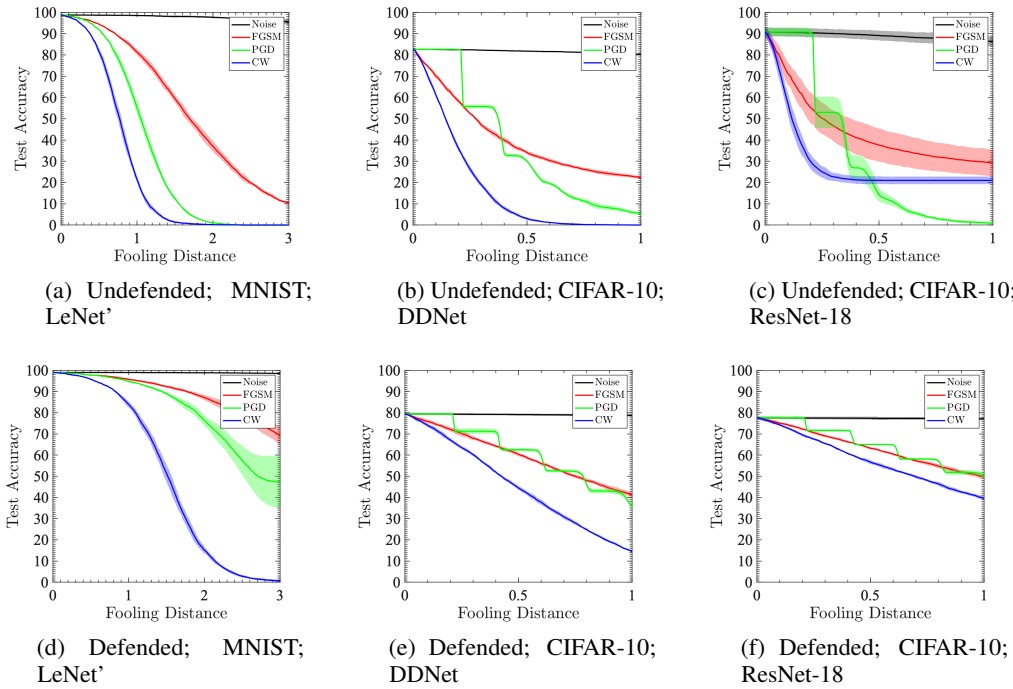

Figure S5: **Effects on test accuracy incurred by various modes of attacks.** (a,d) LeNet' on MNIST, (b,e) DDNet on CIFAR-10, and (c,f) ResNet-18 on CIFAR-10 trained (a,b,c) without defense and (d,e,f) with defense – Jacobian regularization magnitude $\lambda_{\text{JR}} = 0.01$ and adversarial training with $\varepsilon_{\text{FGSM}} \in [0, 0.01]$ – all also include $L^2$ regularization $\lambda_{\text{WD}} = 0.0005$ and (except ResNet-18) dropout rate $0.5$.

## F WHITE NOISE VS. FGSM VS. PGD VS. CW

In Figure S5, we compare the effects of various input perturbations on changing model's decision. For each attack method, fooling $L^2$ distance in the original input space – before preprocessing – is measured between the original image and the fooling image as follows (for all attacks, cropping is performed to put pixels in the range $[0, 1]$ in the orignal space): (i) for the white noise attack, a random direction in the input space is chosen and the magnitude of the noise is cranked up until the model yields wrong prediction; (ii) for the FGSM attack, the gradient is computed at a clean sample and then the magnitude $\varepsilon_{\text{FGSM}}$ is cranked up until the model is fooled; (iii) for the PGD attack, the attack step with $\varepsilon_{\text{FGSM}} = 1/255$ is iterated until the model is fooled [as is customary for PGD and described in the main text, there is saturation constraint that demands each pixel value to be within $32/255$ (MNIST) and $16/255$ (CIFAR-10) away from the original clean value]; and (iv) the CW attack halts when fooling is deemed successful. Here, for the CW attack (see Carlini and Wagner (2017) for details of the algorithm) the Adam optimizer on the logits loss (their $f_6$) is used with the learning rate $0.005$, and the initial value of the conjugate variable, $c$, is set to be $0.01$ and binary-searched for $10$ iterations. For each model and attack method, the shortest distance is evaluated for $1,000$ test samples, and the test error ($= 100\% - \text{test accuracy}$) at a given distance indicates the amount of test examples misclassified with the fooling distance below that given distance.

Below, we highlight various notable features.

- The most important highlight is that, in terms of effectiveness of attacks, $\text{CW} > \text{PGD} > \text{FGSM} > \text{white noise}$, duly respecting the complexity of the search methods for finding adversarial examples. Compared to CW attack, the simple methods such as FGSM and PGD attacks could sometime yield erroneous picture for the geometry of the decision cells, especially regarding the closest decision boundary.

- The kink for PGD attack in Figure S5d is due to imposing saturation constraint that demands each pixel value to be within $32/255$ away from the original clean value. We think that this constraint is unnatural, and impose it here only because it is customary.

- While the CW attack fools almost all the examples for LeNet' on MNIST and DDNet on CIFAR-10, it fails to fool some examples for ResNet-18 on CIFAR-10 (and later on ImageNet: see Section H) beyond some distance. We have not carefully tuned the hyperparameters for CW attacks to resolve this issue in this paper.

## G    DEPENDENCE ON JACOBIAN REGULARIZATION MAGNITUDE

In this appendix, we consider the dependence of our robustness measures on the Jacobian regularization magnitude, $\lambda_{\mathrm{JR}}$. These experiments are shown in Figure S6. Cranking up the magnitude of Jacobian regularization, $\lambda_{\mathrm{JR}}$, generally increases the robustness of the model, with varying degree of degradation in performance on clean samples. Typically, we can double the fooling distance without seeing much degradation. This means that in practice modelers using Jacobian regularization can determine the appropriate tradeoff between clean accuracy and robustness to input perturbations for their particular use case. If some expectation for the amount of noises the model might encounter is available, this can very naturally inform the choice of the hyperparameter $\lambda_{\mathrm{JR}}$.

## H    RESULTS FOR IMAGENET

ImageNet (Deng et al., 2009) is a large-scale image dataset. We use the ILSVRC challenge dataset (Russakovsky et al., 2015), which contains images each with a corresponding label classified into one of thousand object categories. Models are trained on the training set and performance is reported on the validation set. Data are preprocessed through subtracting the $\mathrm{mean} = [0.485, 0.456, 0.406]$ and dividing by the standard deviation, $\mathrm{std} = [0.229, 0.224, 0.225]$, and at training time, this preprocessing is further followed by random resize crop to 224-by-224 and random horizontal flip.

ResNet-18 (see Appendix D) is then trained on the ImageNet dataset through SGD with mini-batch size $|\mathcal{B}| = 256$, momentum $\rho = 0.9$, weight decay $\lambda_{\mathrm{WD}} = 0.0001$, and initial learning rate $\eta_0 = 0.1$, quenched ten-fold every 30 epoch, and we evaluate the model for robusness at the end of 100 epochs. Our supervised loss equals the standard cross-entropy with one-hot targets, augmented with the Jacobian regularizer with $\lambda_{\mathrm{JR}} = 0, 0.0001, 0.0003$, and $0.001$.

Preliminary results are reported in Figure S7. As is customary, the PGD attack iterates FGSM with $\varepsilon_{\mathrm{FGSM}} = 1/255$ and has a saturation constraint that demands each pixel is within $16/255$ of its original value; the CW attack hyperparameter is same as before and was not fine-tuned; $[0, 1]$-cropping is performed as usual, but as if preprocessing were performed with RGB-uniform mean shift $0.4490$ and standard deviation division $0.2260$. The Jacobian regularizer again confers robustness to the model, especially against adversarial attacks. Surprisingly, there is no visible improvement in regard to white-noise perturbations. We hypothesize that this is because the model is already strong against such perturbations even without the Jacobian regularizer, but it remains to be investigated further.

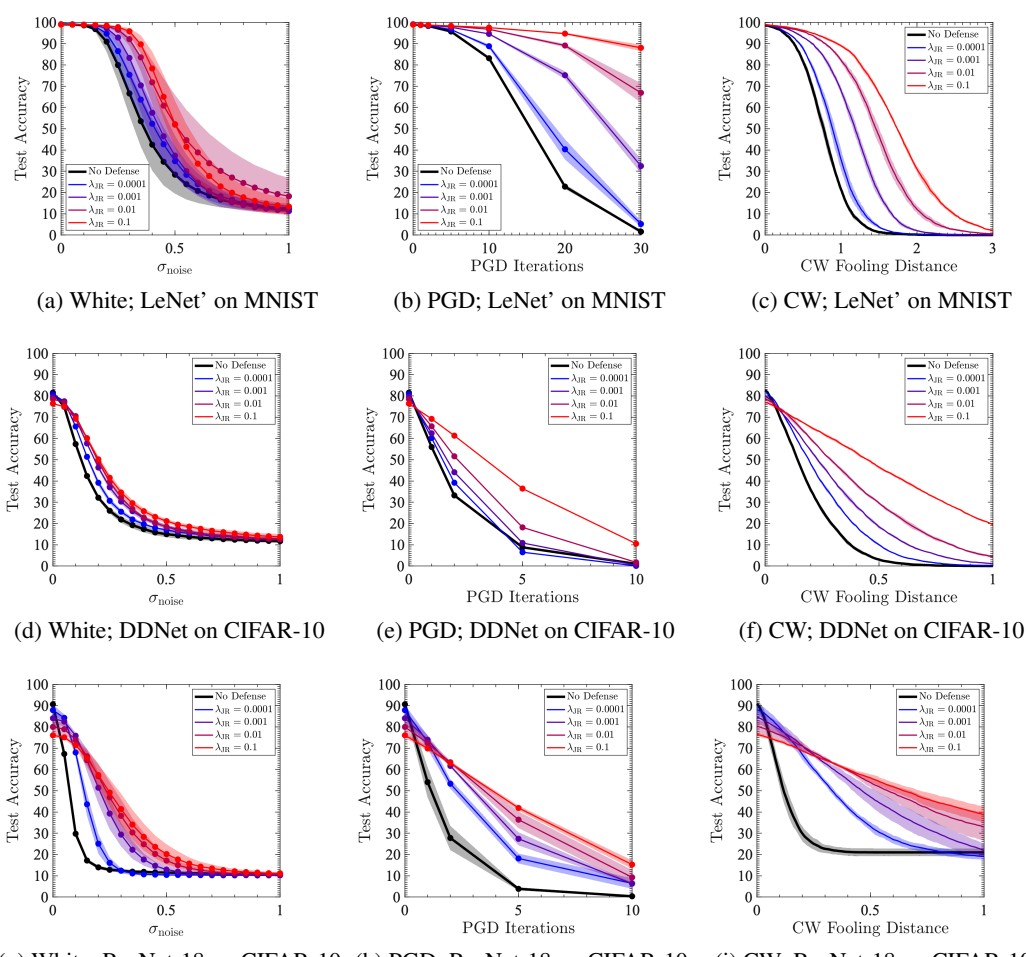

Figure S6: **Dependence of robustness on the Jacobian regularization magnitude $\lambda_{\mathrm{JR}}$.** Accuracy under corruption of input test data are evaluated for various models [base models all include $L^2$ ($\lambda_{\mathrm{WD}} = 0.0005$) regularization and, except for ResNet-18, dropout (rate 0.5) regularization]. Shades indicate standard deviations estimated over 5 distinct runs.

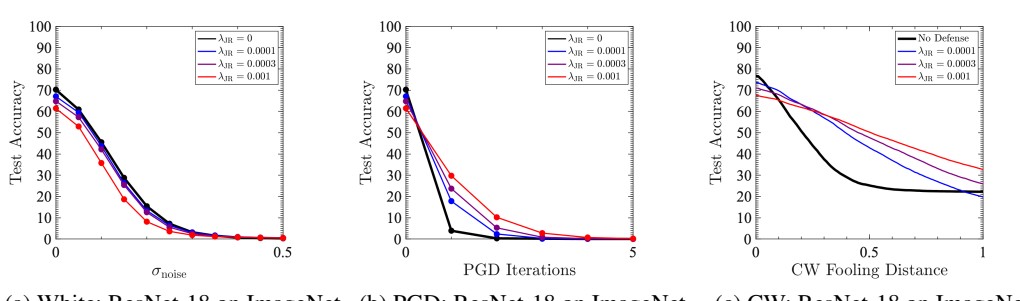

Figure S7: **Dependence of robustness on the Jacobian regularization magnitude $\lambda_{\mathrm{JR}}$ for ImageNet.** Accuracy under corruption of input test data are evaluated for ResNet-18 trained on ImageNet [base models include $L^2$ ($\lambda_{\mathrm{WD}} = 0.0001$)] for a single run. For CW attack in (c), we used 10,000 test examples (rather than 1,000 used for other figures) to compensate for the lack of multiple runs.

