# OpenReview forum: "Robust Learning with Jacobian Regularization"
_ICLR.cc/2020/Conference — Reject_

### Official Review · AnonReviewer1 · 2019-10-22
**Official Blind Review #1**

**Rating:** 6

**Review:**

Summary:

Stability is one of the important aspects of machine learning.  This paper views Jacobian regularization as a scheme to improve the stability, and studies the behavior of Jacobian regularization under random input perturbations, adversarial input perturbations, train/test distribution shift, and simply as a regularization tool for the classical setting without any distribution shifts nor perturbations.  There are already several related works that propose to use Jacobian regularization, but previous works didn’t have an efficient algorithm and also did not have theoretical convergence guarantee.  This paper offers a solution that efficiently approximate the Frobenius norm of the Jacobian and also show the optimal convergence rate for the proposed method.  Various experiments show that the behavior of Jacobian regularization and show that it is robust.

Reasons for the decision:

Positives: The contribution of the paper seems to be two-fold:  First a theoretically guaranteed and efficient method for Jacobian regularizer, and second, intensive experiments to show the robustness of the Jacobian regularizer.  Each of these points seem to have important contributions for the field.

Negatives:  An issue might be that the latter contribution seems to be orthogonal to the former since there are no experiments comparing with previous methods mentioned in the paper, and it gives the impression that there are two separate stories in one paper.  For instance, there are no experiments comparing computational time between Sokolic et al. (2017). On the other hand, there are only regularization methods (that are not necessarily designed to be robust) used as baselines in the experiments to show robustness, instead of algorithms that are designed to be robust, e.g., domain adaptation methods for Table 2.  It would make the paper stronger to combine these two lines of contributions into a single story.  For example, it might be better to emphasize more that experiments such as Figure S7 was previously not possible due to inefficient implementation.

Minor comment:  It would make the paper stronger to include some of the main related works in the Introduction section.


After author response:
Thank you for reading my review and answering the questions.  Although I still feel the same for my score (6), in my opinion, the same issues exist for this paper, and the paper can be made stronger on those points.

**Experience Assessment:**

I have read many papers in this area.

**Review Assessment: Checking Correctness Of Derivations And Theory:**

I did not assess the derivations or theory.

**Review Assessment: Checking Correctness Of Experiments:**

I carefully checked the experiments.

**Review Assessment: Thoroughness In Paper Reading:**

I read the paper at least twice and used my best judgement in assessing the paper.

---

> ### Author Response · Authors · 2019-11-15
> **Review Response**
>
> Thank you for your feedback on our work. We agree with your assessment of our main contributions and also appreciate your suggestions for further improvements. We will take your suggestion to further emphasize the difficulty of producing results like those shown in Fig S7 without an efficient approximate algorithm.
>
> We agree that the L2 and dropout regularizers are not intended to provide adversarial robustness. However, Jacobian regularization is similarly a generic regularizer which prioritizes input-output stability. We believed that a natural comparison between commonly used regularizers to prevent overfitting in deep learning were a useful initial comparison point. In the majority of our experiments we additionally include a comparison with adversarial training which is specifically designed for defense against the types of attacks we use at test time. We highlight that Jacobian regularizer is able to improve robustness on par with a method that privileged information (i.e. knowledge of likely test time attacks during learning). This showcases the general applicability of Jacobian regularizer and makes the ability to scale the approach to large data settings all the more relevant.
>
> We also want to clarify that our method when applied for a scenario like cross-domain generalization is trained using *only* access to the source domain and is simply applied at test time to target domain. This does not invalidate the potential use of adaptation approaches in addition to Jacobian regularization should (potentially unlabeled) target data become available during training. Our goal is to showcase the intrinsic robustness to data shift that arises from optimizing Jacobian regularizer.

---

### Official Review · AnonReviewer2 · 2019-10-23
**Official Blind Review #2**

**Rating:** 3

**Review:**

This paper proposes an efficient method to (differentiably) estimate input-output Jacobian. The method is useful for Jacobian regularization. The regularization improves robustness and generalization of networks.

I tend to vote for rejection. There are two concerns. 1) This paper needs to demonstrate the effectiveness of the input-output Jacobian regularization over the input gradients regularization. 2) It is doubtful whether the regularizer provides the same benefits mentioned in Experiment 3.1 for other datasets than MNIST.

Major comments:
1) This paper needs to demonstrate applications that the regularization of input-output Jacobian is more beneficial than that of input gradients. Input gradients regularization has repeatedly appeared in the literature, as the paper mentions. For example, [1] regularized input gradients to improve the robustness against adversarial examples. The input gradients regularization is computationally more efficient than Varga et al. (2017), with which the submitted paper compares the proposed method. If input gradients regularization is sufficient, it limits the impact of the submitted paper. It is strongly encouraged to demonstrate when and why the input-output Jacobian regularization is preferable.
2)  Experimental results on CIFAR10 and ImageNet show accuracy degradation on clean test data. It is questionable whether we can reach the same conclusion with the experiments 3.1 on those datasets.

[1] Andrew Slavin Ross and Finale Doshi-Velez. "Improving the Adversarial Robustness and Interpretability of Deep Neural Networks by Regularizing their Input Gradients." AAAI 2018

Update =====

Thank you for the authors' response. I agree that the regularization of the input-output Jacobian is potentially superior to double back-prop. However, as authors are aware of, it needs to be validated experimentally. I think this paper should not leave this as a future work, and hence keep the review score.

**Experience Assessment:**

I have published one or two papers in this area.

**Review Assessment: Checking Correctness Of Derivations And Theory:**

I assessed the sensibility of the derivations and theory.

**Review Assessment: Checking Correctness Of Experiments:**

I assessed the sensibility of the experiments.

**Review Assessment: Thoroughness In Paper Reading:**

I read the paper at least twice and used my best judgement in assessing the paper.

---

> ### Author Response · Authors · 2019-11-15
> **Review Response**
>
> Thank you for your comments and valuable feedback. We agree that comparison against DoubleBackProp (or the variant by Ross et al.) would be interesting. Note, however, that we were inspired by the observation from the 2017 IEEE Security and Privacy paper from Carlini & Wagner which compared a set of potential attacks and found that the strongest attack operated on bare logits instead of softmax values or loss functions (See Table III from their paper and superiority of f_6). As our goal is to enforce robustness against any attack we believe there is value in minimizing the norm of the input-output Jacobian matrix directly instead of minimizing aggregate loss perturbations. Also note that we do not know a priori which class will be used by adversaries as a fooling target and thus it is best to minimize changes of outputs for all classes.  Of course, this conjecture needs to be further experimentally validated and we will consider this in future work.
>
> Regarding the clean accuracy drop on CIFAR-10 and ImageNet, there is often a fundamental tradeoff between robustness and clean performance which is well documented across the adversarial robustness literature. In our case, we have the ability to tune the loss weight on the regularizer to choose between more robustness or higher clean performance -- this is demonstrated in Figure S7. Note that as the Jacobian loss weight is set smaller the resulting model still retains consistently improved robustness over a clean model across multiple attacks, while minimizing the degradation in clean accuracy.

---

### Official Review · AnonReviewer3 · 2019-10-24
**Official Blind Review #3**

**Rating:** 3

**Review:**


The main contribution of this paper is that it proposed an estimator of Jacobian regularization term for neural networks to reduce the computational cost reduced by orders of magnitude, and the estimator is mathematically proved unbiased. In details, the time consumed for the application of Jacobian regularizer and the unbiasedness of the proposed estimator are proved mathematically. Then the author experimentally demonstrated that the proposed regularization term retains all the practical benefits of the exact method but with a low computation cost. Quantitative experiments are provided to illustrate that the proposed Jacobian regularizer does not adversely affect the model, can be used simultaneously with other regularizers and effectively improve the model's robustness against random and adversarial input perturbations.

In general, this paper was well organized and it is also great that an efficient approximation of the Jacobian regularizer can be derived. However, the paper was written in a quite misleading or over-claimed way. The major comments are as follows:

(1)	It is not new to use Jacobian regularizer for improving the robustness learning. Such idea has been elaborated in [1] (though it was cited in the paper). The main contribution is, the paper proposed an efficient approximated way to the exact Jacobian term. All the benefits of robust learning are rooted in the Jacobian regularization.
(2)	In light of point (1), the title of this paper was first quite misleading. Robust learning with Jacobian was not proposed by the paper, so the title makes no sense.  Instead, efficient approximation should be emphasized.
(3)	Most of the advantages shown in the quantitative experiments are the benefits of Jacobian regularization! The authors should focus on the approximating algorithm rather than the merits of Jacobian regularization which has been discussed in [1]. In another word, it is better to add more comparison based on running time so as to illustrate the significant performance between the proposed regularizer and the exact one.
(4)	In section 3, it may be interesting to see more comparison results if two regularizers were combined, e.g., (L2+Dropout, L2+Jacobianor Dropout+Jacobian).

[1] Varga, Dániel, Adrián Csiszárik, and Zsolt Zombori. "Gradient regularization improves accuracy of discriminative models." arXiv preprint arXiv:1712.09936 (2017).


===============
I have read the authors' response. I would have to say that, though I enjoy reading the paper and the other parts of this paper are very good,  the paper's contribution/novelty may be limited because Jacobian regularization has been earlier thoroughly discussed in [1] and even another recent  paper published in ECCV 2018.

I am afraid that I  would firmly keep my original rating.

**Experience Assessment:**

I have published in this field for several years.

**Review Assessment: Checking Correctness Of Derivations And Theory:**

I assessed the sensibility of the derivations and theory.

**Review Assessment: Checking Correctness Of Experiments:**

I carefully checked the experiments.

**Review Assessment: Thoroughness In Paper Reading:**

I read the paper thoroughly.

---

> ### Author Response · Authors · 2019-11-15
> **Review Response**
>
> Thank you for your comments. The goals of our work (as correctly summarized by R3) are as follows: we argue that Jacobian regularizer is useful for producing robust representations. While others have made similar claims, the previous work lacked the empirical evidence and efficient implementation required to scale to the variety of settings tackled in modern visual recognition systems. We present an efficient approximation to Jacobian regularizer. We first empirically verify that it is a tight approximation of the exact solution (see Fig 2). Our key contribution is then two-fold: (i) our approximation *enables* the potential for using Jacobian regularization even as the number of recognition classes scales and (ii) we provide extensive experimental evidence for the usefulness of Jacobian regularizer across a variety of architectures, datasets, and number of categories.
>
> We argue that the efficient approximate computational scheme, together with thorough experiments applied to a variety of settings which showcase the general value of the Jacobian regularizer, produces a potentially impactful contribution over prior work. We believe that our novel experimental justification contributes evidence to the scientific community and that our efficient implementation with associated released code will enable easy adoption of this regularizer.
>
> Response to Specific Comments
> =========================
> (2) We appreciate your feedback on the title of our work and strive for clarity in our presentation. We propose the following alternative title: “Efficient Jacobian Regularization for Scalable Robust Learning.”
> (3) Indeed, most experiments center along extensive experiments showcasing the regularization benefits that can be achieved with our efficient approximate Jacobian regularizer.
> (4) We would be happy to include a combination of two regularizers into Section 3. In fact, we experimented with this originally but did not find any meaningful difference between this result and what is presented in Table 1. The main message from Table 1 is to say that although each regularizer performs comparably when applied independently, there is an additional benefit from using the combination of regularizers. We believe that this is a nontrivial and important point to justify the addition of Jacobian regularizer into modern network training procedures which frequently use L2 weight decay and/or dropout.

---

### Decision · Program_Chairs · 2019-12-19

**Decision:**

Reject

**Comment:**

Three reviewers have reviewed this submission and scored it as 6/3/3. After rebuttal, the reviewers remained unconvinced. The main criticisms concerns the Jacobian  regularizaton [1] being known which makes the contributions of this submission  look diluted. Additionally, there were concerns over results (degradation) on CIFAR10 and ImageNet and other minor issues.
For these reasons, this paper cannot be accepted by ICLR2020.